# Molecular Characteristics and Antimicrobial Resistance of *Salmonella enterica* Serovar Schwarzengrund from Chicken Meat in Japan

**DOI:** 10.3390/antibiotics10111336

**Published:** 2021-11-01

**Authors:** Kaoru Matsui, Chisato Nakazawa, Shwe Thiri Maung Maung Khin, Eriko Iwabuchi, Tetsuo Asai, Kanako Ishihara

**Affiliations:** 1Laboratory of Veterinary Public Health, Faculty of Agriculture, Tokyo University of Agriculture and Technology, 3-5-8 Saiwai-cho, Fuchu 183-8509, Tokyo, Japan; matsuika1341@gmail.com (K.M.); chisato.nakazawa.0218@gmail.com (C.N.); s213944v@st.go.tuat.ac.jp (S.T.M.M.K.); 2Japan Veterinary Medical Association, 1-1-1 Minami-Aoyama, Minato City 107-0062, Tokyo, Japan; 3Department of Nutrition, School of Nursing and Nutrition, Tenshi College, Kita 13 Higashi 3, Higashi-ku, Sapporo 065-0013, Hokkaido, Japan; iwabuti@tenshi.ac.jp; 4United Graduate School of Veterinary Science, Gifu University, 1-1 Yanagido, Gifu City 501-1193, Gifu, Japan; tasai@gifu-u.ac.jp

**Keywords:** class 1 integron, foodborne pathogen, multilocus sequence typing, *Salmonella enterica* serovar Schwarzengrund

## Abstract

Our previous study revealed that *Salmonella enterica* serovar Schwarzengrund-contaminated areas of broiler chickens have expanded from West Japan to East Japan. The present study investigated the antimicrobial resistance and molecular characteristics of 124 *S.* Schwarzengrund isolates obtained from chicken meat produced in East and West Japan from 2008 to 2019. Comparing the isolates obtained in 2008 and 2015–2019, an increase in the proportion of those resistant to kanamycin [51.4–89.7% (*p* < 0.001)] was observed. In contrast, the proportion of isolates resistant to both streptomycin and tetracycline and those that harbored a 1.0-kb class 1 integron, *aadA1*, and *tetA*, significantly decreased from 100% in 2008 to 47.1% in 2015–2019 (*p* < 0.001). A 1.0-kb class 1 integron containing *aadA1*, harbored by 78 isolates, was different from that reported in globally distributed *S.* Schwarzengrund strains (1.9 kb, containing the *dfrA12-aadA2* gene cassette). Twenty-five isolates from different product districts and years of isolation were typed as sequence type (ST) 241 with multilocus sequence typing. Our results suggest that *S.* Schwarzengrund, which contaminates chicken meat in Japan, shares a common ancestor regardless of the product district from 2008 to recent years. Moreover, *S.* Schwarzengrund ST241 may have spread from western to eastern Japan.

## 1. Introduction

*Salmonella* is one of the main causes of bacterial gastroenteritis worldwide and can colonize the intestinal tracts of food animals [1]. Most patients with salmonellosis are infected through ingestion of contaminated animal products. Meat, especially poultry and pig meat, is the second most common cause of salmonellosis [2,3]. In particular, chicken meat is frequently associated with human salmonellosis [4]. Approximately half of all retail chicken meat produced in Japan in 2011–2015 was contaminated with *Salmonella* [5,6]. According to the Ministry of Agriculture, Forestry, and Fisheries, the supply of broiler chickens for domestic consumption has increased by 26% to reach 2.5 million tons over the last 10 years (2009–2019); this was the highest rate of increase among all types of meat in Japan [7]. As the consumption of chicken meat is increasing, exposure of humans to *Salmonella* through chicken meat would be more frequent than before.

*Salmonella enterica* serovar Schwarzengrund has been isolated from chicken meat and human patients in Thailand [8] and Taiwan [9], as well as from meat and fecal samples of chickens in Brazil [10]. In Korea, *S.* Schwarzengrund was isolated from cattle between 2010 and 2012 [11]. The characteristics were almost the same among *S.* Schwarzengrund isolates from human patients and those from chicken meat or sick animals [12,13]. Therefore, it was considered that *S.* Schwarzengrund, which colonized or infected food animals, was transmitted to humans through food. The control of *S*. Schwarzengrund-contaminated foods is important for the prevention of human salmonellosis.

In Japan, the proportion of *S.* Schwarzengrund isolates from broiler chickens in farms was low (1.3% in 1999 [14] and 0% in 2000–2003 [15]). However, this proportion increased to 28.1% between 2005 and 2007 [16]. The proportion of *S*. Schwarzengrund isolated from broiler chickens at poultry processing plants in Kagoshima, Japan also increased from 2.1% in 2009–2012 [17] to 21.3% in 2013–2016 [18]. Furthermore, the distribution of *S*. Schwarzengrund has expanded in Japan [19]. It is currently unknown whether *S.* Schwarzengrund in West Japan has expanded to East Japan or whether new *S.* Schwarzengrund clones from other countries have invaded East Japan. Among *Salmonella*, except *S*. *enterica* serovar Typhi and *S*. *enterica* serovar Paratyphi A from human patients, the percentage of *S*. Schwarzengrund increased from 1.6% (89/5488) in 2006–2010 [20] to 9.9% (162/1633) in 2017–2021 [21].

To determine the origin of *S.* Schwarzengrund in East Japan, isolates from retail chicken meat produced in Japan were characterized in terms of antimicrobial resistance and genotypes.

## 2. Results

### 2.1. Antimicrobial Susceptibility

The total number and percentages of antimicrobial-resistant isolates are shown in Table 1. Although antimicrobial susceptibility in ampicillin (AMP), cefazolin (CFZ), streptomycin (STR), gentamicin (GEN), kanamycin (KAN), chloramphenicol (CHL), nalidixic acid (NAL), and trimethoprim (TMP) for 29 isolates in 2008 was reported previously [22], the MICs of tetracycline (TET), ciprofloxacin (CIP), and trimethoprim-sulfamethoxazole (SXT) for the isolates were determined in this study. The minimum inhibitory concentrations (MICs) of all 11 antimicrobials for the remaining eight isolates from 2008 and 87 isolates from 2015–2019 were reported in this study. All isolates in 2008 and 2015–2019 were susceptible to AMP, CFZ, GEN, CHL, and CIP. Then, the percentage of resistant isolates was compared between 2008 and 2015–2019. The proportion of STR and TET resistance decreased significantly from 100% (37/37) to 47.1% (41/87, *p* < 0.001). The proportion of TMP resistance also decreased significantly from 56.8% (21/37) to 16.1% (14/87, *p* < 0.001). Moreover, the proportion of SXT resistance decreased significantly from 70.3% (26/37) to 44.8% (39/87, *p* = 0.009). In contrast, the proportion of KAN resistance significantly increased from 51.4% (19/37) to 89.7% (78/87, *p* < 0.001). Although the proportion of NAL resistance increased from 0% (0/37) to 9.2% (8/87), a significant difference was not observed (*p* = 0.1038).

### 2.2. Antimicrobial Resistance Genes

Antimicrobial resistance genes were tested in all 124 isolates. Using multiplex polymerase chain reaction (PCR) 1, 2, and 3, *aadA* (78 isolates), *strA*/*strB* (5 isolates), *aphA1* (97 isolates), and *tetA* (78 isolates) were determined. Then, antimicrobial resistance genes of the selected isolates were sequenced. Using Nucleotide BLAST analyses, the 479 bp sequences of *aadA* in Sal_15, Sal_17, and Sal_27 had 100% identity with *aadA1* in *Salmonella enterica* serovar Typhimurium (accession no. MT507883; position, 14,482 to 14,960). The 476 bp sequences of *tetA* in Sal_T1, Sal_17, Sal_100, and Sal_276 had 100% identity with *tetA* in *Salmonella enterica* serovar Infantis (accession no. CP066336; position, 175,555 to 176,030). The 800 bp sequences of *strA*/*strB* in Sal_264 and Sal_270 had 100% (191/191) identity with *strA* (accession no. AP019374; position, 1242600 to 1242790), and 100% (610/610) identity with *strB* (accession no. AP019374; position, 1241991 to 1242600). Lastly, the 605 bp sequences of *aphA1* in Sal_15, Sal_69, Sal_139, and Sal_268 had 100% identity with *aphA1* (*aph (3**′)-Ia*) in *S*. Infantis (accession no. CP052818; position, 300840 to 301444).

Whole genome sequencing (WGS) analysis was performed using five isolates (Sal_167, Sal_249, Sal_266, Sal_278, and Sal_291). In addition to the antimicrobial resistance genes mentioned above, *sul1*, *sul2*, *dfrA14*, and *aac(6′)-Iaa* were detected using ResFinder. A summary of antimicrobial resistance genes and their sequences detected by ResFinder is shown in Appendix A. These genes were also tested for all isolates using multiplex PCR 4, PCR 1, and PCR 2. Then, *sul1* (78 isolates), *sul2* (5 isolates), *dfrA14* (69 isolates), and *aac(6**′)-Iaa* (all 124 isolates) were detected.

Table 2 shows the 14 antimicrobial resistance patterns and eight confirmed antimicrobial resistance gene patterns. Bicozamycin, which was tested in a previous study [22], could not be tested in this study. Therefore, bicozamycin resistance was excluded from the antimicrobial resistance patterns previously reported. The number of antimicrobial resistance patterns increased from six types in 2008 to 12 types in 2015–2019. Furthermore, the number of antimicrobial resistance gene patterns also increased from three patterns in 2008 to eight patterns in 2015–2019. These results indicate that the characteristics of antimicrobial resistance have diversified over time. The detected antimicrobial resistance genes were as follows: STR resistance, *aadA1*; TET resistance, *tetA*; KAN resistance, *aphA1*; TMP resistance, *dfrA14*; sulfonamide resistance, *sul1* or *sul2*.

The antimicrobial resistance genes detected were consistent with the antimicrobial susceptibility phenotypes, with some exceptions. Three out of sixty-nine *dfrA14*-positive isolates (Sal_T36, Sal_G4, and Sal_G43) were susceptible to TMP. The 455 bp sequences of *dfrA14* in Sal_T36, Sal_G4, Sal_G43, Sal_63, Sal_112, Sal_152, Sal_247, and Sal_270 had 100% identity with *dfrA14* in *Escherichia coli* (accession no. CP072323; position, 72653 to 73107). Moreover, five *strA*/*strB*-positive isolates were susceptible to STR (MIC = 4 or 8 μg/mL).

### 2.3. Class 1 Integron Detection and Characterization

The *intI1* was detected in 62.9% of isolates (78/124), and all 78 isolates harbored *aadA*, *tetA*, and *sul1*. Approximately 1.0 kb of gene cassettes of class 1 integron was amplified for all *intI1*-positive isolates. Although all 37 isolates from 2008 harbored a 1.0-kb class 1 integron, *aadA*, *tetA*, and *sul1*, these genes were not detected in 46 out of 87 isolates in 2015–2019 (52.9%). Meanwhile, sixty-one out of sixty-nine *dfrA14*-positive isolates (88.4%) harbored a 1.0-kb class 1 integron. The amplicon of one isolate (Sal_G1) was sequenced, and DNA alignment showed 100% (862/862) identity with the class 1 integron containing the *aadA1* gene cassette (accession no. GU987053; position, 80 to 941). Based on the DNA alignment of Sal_G1, the amplicon was expected to be cleaved into 12, 15, 31, 34, 100, 294, and 523-bp fragments using TaqI. Through agarose gel electrophoresis of digested PCR products for gene cassettes of class 1 integron of Sal_G1 using FastDigest TaqI (Thermo Fisher Scientific K. K., Tokyo, Japan), fragments estimated to be 100, 294, and 523-bp were confirmed. The restriction fragment length polymorphism (RFLP) patterns of the remaining 77 isolates matched those of Sal_G1 (Appendix A).

### 2.4. Multilocus Sequence Type (MLST) Analysis

Twenty isolates were selected from different origins (product districts and years of isolation) and resistance gene patterns for MLST analysis (Appendix A). The nucleotide sequences of these 20 isolates were determined after amplifying seven genes using PCR. All 20 isolates were assigned to the sequence type (ST) 241 isolates. In addition, the *hisD* sequences of the remaining 104 isolates were determined. The *hisD* was identical to allele type 16, which is the same allele type as ST241, for all isolates. Moreover, five isolates analyzed using WGS were also assigned to ST241 (Appendix A).

Data from the registered *Salmonella enterica* serovar Schwarzengrund on 12 May 2020 were obtained from Salmonella Genome Databases [23] and EnteroBase [24] and were combined into one dataset. Duplicate isolate data were deleted using the duplicate deletion function of Excel 2016 (Microsoft Corporation, WA, USA). In the two databases, a total of 1707 *S.* Schwarzengrund isolates were registered and assigned to 35 types. Appendix A shows the geographical distribution of STs of the 1707 isolates. ST241, to which 25 isolates obtained from chicken meat were assigned in this study, was reported mainly in the UK and USA, where ST96 was the dominant type (Appendix A).

In addition, ST96 [1609/1707 (94.3%); allelic profile, 43-47-49-49-41-15-3] was the most registered type, followed by ST241 [30/1707 (1.8%); 43-47-49-16-41-15-3], ST2250 [14/1707 (0.8%); 516-13-401-131-522-2-531], and ST322 [12/1707 (0.7%); 43-47-49-49-41-15-114]. Nine out of thirty-five STs (namely, ST241, ST322, ST5010, ST2114, ST7288, ST848, ST5405, ST6132, and ST7884) were ST96-related STs that shared five or six of their loci with ST96. A seven-base difference in *hisD* was observed in ST241 compared with that in ST96, whereas a one-base difference in each locus was observed in other types in the ST96-related group (Appendix A). As an exception, ST2114 differed by one and three bases in *hisD* and *thrA*, respectively. The one-base difference in the *hisD* of ST2114 was included in the seven-base difference in ST241. In contrast, 25 types differed from ST96 in all seven loci (Appendix A).

An improved minimal spanning tree based on 1833 *S.* Schwarzengrund isolates that were typed using MLST and were registered in EnteroBase [24] on 5 March 2021, were generated using GrapeTree within EnteroBase [24] (Figure 1). When GrapeTree was used to generate the figure, an algorithm called MSTree V2 was selected. The node size and kurtosis were set to 300% and 30%, respectively.

## 3. Discussion

The present study showed that *Salmonella enterica* serovar Schwarzengrund isolates from chicken meat produced in West and East Japan were typed as ST241. In addition, even though multiple antimicrobial resistance patterns of the isolates were observed, the diversity of antimicrobial resistance genes was limited. Our results suggest that the *S*. Schwarzengrund isolates that contaminate chicken meat in Japan share a common ancestor regardless of their antimicrobial resistance pattern, the product district, and the year of isolation. *S.* Schwarzengrund ST241 may have spread from western to eastern Japan.

For the discrimination of *S.* Schwarzengrund, we utilized MLST, which involves seven housekeeping genes [25]. Previously, this MLST approach was successfully used to discriminate *Salmonella enterica* serovar Typhimurium from its four serologically close relatives [26]. The MLST approach was also applied to *S.* Schwarzengrund. Isolates obtained in the UK in 1988 [27], Taiwan, and Denmark [28] were typed as ST96. In another study, 122 isolates from fecal samples from broiler farms in Kyushu, West Japan from 2013 to 2014 were typed as ST241 [29]. The search in Salmonella Genome Databases [23] and EnteroBase [24] conducted in the present study showed that 1707 *S.* Schwarzengrund isolates were registered and classified into 35 types using this MLST approach, with 94.3% of the isolates typed to ST96. All seven allele numbers for 25 of the 35 STs differed from those of ST96. Only three groups consisting of related STs [(1) ST96 and nine types related to ST96, (2) ST2250 and ST5815, and (3) ST19 and ST213] differed only with regard to one or two loci. Twenty-one types not included in these three groups were unrelated to each other. These results suggest that the MLST approach discriminated different ancestors of *S.* Schwarzengrund.

In the present study, six isolates obtained from chicken meat produced in West Japan in 2008 and 19 isolates obtained from chicken meat produced in West and East Japan between 2015 and 2019 were selected for MLST. All 25 isolates were typed as ST241. The sequence of *hisD*, a housekeeping gene used for MLST, in all the remaining 99 isolates also matched with that of ST241. Therefore, none of the 124 isolates were identified as ST96. This study revealed the molecular characteristics of *S*. Schwarzengrund isolated from a larger area in older and more recent years than in a previous study [29]. These data suggest that *S*. Schwarzengrund-contaminated broiler flocks and chicken meat in Japan share a common ancestor, regardless of the antimicrobial resistance patterns, product districts, and years of isolation. As *S.* Schwarzengrund was found only in West Japan prior to 2012 [5,22,30], *S.* Schwarzengrund ST241 may have spread from West to East Japan in 2013 or later.

Upon confirming the differences in sequences of seven housekeeping genes between ST96 and ST96-related STs, including ST241, no intermediate variants between ST96 and ST241, which differ from ST96 by one to six bases in *hisD*, were registered in the databases. In the MLST approach, the contribution of recombination to both clonal divergence and point mutations was considered [31]. Therefore, ST241 could be derived from ST96 through recombination in the *hisD* region. Alternatively, intermediate variants may be present among unregistered *S*. Schwarzengrund isolates.

A previous study reported that multidrug-resistant *S.* Schwarzengrund exhibiting resistance to AMP, GEN, CHL, CIP, NAL, STR, sulfamethoxazole, and TET has spread to Denmark and the USA by imported chicken products from Thailand [12]. However, none of the isolates in the present study were resistant to AMP, GEN, CHL, or CIP. Moreover, a 1.9-kb class 1 integron containing the *dfrA12-aadA2* gene cassette was detected in isolates from Taiwan [32], Thailand [33], Germany [34], and Denmark [26], suggesting that this integron is geographically widespread among *S.* Schwarzengrund. In contrast, isolates in our study harbored a 1.0-kb class 1 integron containing *aadA1*. Therefore, *S.* Schwarzengrund in Japan is suggested to differ from the globally distributed *S.* Schwarzengrund based on the characteristics of antimicrobial susceptibilities and class 1 integrons.

A significant decrease (STR, TET, TMP, and SXT) and increase (KAN) in the antimicrobial resistance rate of *S.* Schwarzengrund between 2008 and 2015–2019 were observed in this study. The correlation between veterinary antimicrobial use and antimicrobial resistance of *Escherichia coli* in food animals has been reported in Japan [35] and seven EU countries [36]. We then confirmed the amount of pure active substances in annual reports from the National Veterinary Assay Laboratory [37]. For broiler chickens in Japan, the sales amount of oxytetracycline decreased from 12,237.1 kg in 2008 to 6211.9 kg in 2016. The sales amount of STR or dihydrostreptomycin (5803.5–8223.3 kg), that of KAN (1145.2–3571.3 kg), and that of TMP (247.0–1173.2 kg) for broiler chickens increased from 2008 to 2016.

As *S.* Schwarzengrund isolates were classified as resistant or susceptible to both STR and TET, the use of either antimicrobial could select isolates resistant to them. The significant decrease in the proportion of resistance to both STR and TET in *S.* Schwarzengrund may have occurred because of the considerable reduction in oxytetracycline sales. The increase in KAN use was thought to contribute to this significant increase in resistance among *S.* Schwarzengrund. Although the sales volume of TMP increased, the proportion of TMP resistance in *S.* Schwarzengrund decreased significantly. The factor that decreased TMP-resistant *S.* Schwarzengrund could not be identified.

## 4. Materials and Methods

### 4.1. Bacterial Isolates

A total of 124 *Salmonella enterica* serovar Schwarzengrund isolates from retail chicken meat were tested in this study (Appendix A). In previous studies, 29 and 56 isolates were obtained from samples produced in 2008 [22] and 2015–2017 [19], respectively. In addition, this study included 8, 20, and 11 isolates from retail chicken meat produced in 2008, 2017, and 2019, respectively.

Of the 124 isolates, 47 and 49 isolates were obtained from samples produced in West and East Japan, respectively. The product districts for samples from which the remaining 28 isolates were obtained could not be identified; thus, these samples were labeled as ‘domestic’ (Appendix A). One isolate from each sample was selected for this study, with the exception of two samples produced in 2019, from which two isolates each were obtained as they showed different antimicrobial resistance patterns.

### 4.2. Antimicrobial Susceptibility Test

The MICs of the 11 antimicrobial agents against 8 and 87 *S.* Schwarzengrund isolates obtained from chicken meat produced in 2008 and 2015–2019, respectively, were determined. These 11 antimicrobials were selected according to the Japanese Veterinary Antimicrobial Resistance Monitoring System [37]. The MICs of nine antimicrobial agents [AMP (breakpoint, 32 μg/mL), CFZ (8 μg/mL), GEN (16 μg/mL), KAN (64 μg/mL), TET (16 μg/mL), CHL (32 μg/mL), NAL (32 μg/mL), CIP (1 μg/mL), and SXT (4/76 μg/mL)] were determined with the broth micro-dilution method using Frozen Plate (Eiken Chemical Co., Ltd., Tokyo, Japan). In addition, MICs for STR (16 μg/mL) (FUJIFILM Wako Pure Chemical Co., Osaka, Japan) and TMP (16 μg/mL) (FUJIFILM Wako Pure Chemical Co.) were determined using the agar dilution method in Mueller–Hinton agar (Thermo Fisher Scientific K. K.), according to the Clinical Laboratory Standards Institute (CLSI) guidelines [38]. Of the above-mentioned agents, the MICs of eight antimicrobial agents, excluding TET, CIP and SXT, were determined for 29 of 37 isolates obtained from samples produced in 2008 in a previous study [22]. In that study, the MICs of dihydrostreptomycin instead of STR were determined. The results of the previously reported antimicrobial susceptibility tests were also used in this study. The MICs for TET, CIP, and SXT for the 29 isolates were determined in this study.

*Enterococcus faecalis* ATCC29212, *Staphylococcus aureus* ATCC29213, and *E. coli* ATCC25922 were used for quality control. The breakpoints reported by the CLSI guidelines [38] were applied, with the exception of breakpoints for STR. An intermediate MIC of bimodal distribution was determined as the breakpoint for STR in this study.

### 4.3. Antimicrobial Resistance Gene Detection

The bacterial isolates were grown overnight at 35 °C on Mueller–Hinton agar. The culture was suspended in DNAzol Direct (Molecular Research Center, Inc., Cincinnati, OH, USA) and heated at 95 °C for 15 min. These lysates were used as the DNA templates. The major genes for resistance to STR (*aadA* and *strA*/*strB*; Multiplex PCR 1), KAN (*aphA1*, and *aphA2*; Multiplex PCR 2), TET (*tetA* and *tetB*; Multiplex PCR 3) [39] were tested with multiplex PCR using Go Taq Green Master Mix (Promega K. K., Tokyo, Japan) with minor modifications. The primer sequences, expected PCR product sizes, PCR conditions, positive control, and references are listed in Appendix A. The PCR products of each antimicrobial resistance gene of representative isolates were purified using the Favorprep GEL/PCR Purification Mini Kit (FAVORGEN Biotech Corporation, Ping-Tung, Taiwan) and sequenced by FASMAC Co. (Kanagawa, Japan) or the Division of Genomics Research, Life Science Research Center, Gifu University (Gifu, Japan). The Nucleotide BLAST program was used to compare DNA alignments with data from the National Center for Biotechnology Information using GENETYX version 14 (Genetyx Corporation, Tokyo, Japan).

### 4.4. Integron Analysis

To determine the inserted gene cassettes of the class 1 integron, the regions of each isolate were classified using RFLP. First of all, 124 isolates were tested for the class 1 integrase gene (*intI1*; PCR 3 in Appendix A) [40] via PCR using Go Taq Green Master Mix (Promega K. K.). Furthermore, the gene cassettes of the class 1 integron inserted in all *intI1*-positive isolates were amplified with PCR 4 in Appendix A [40] using the TaKaRa Ex Taq DNA Polymerase (Takara Bio Inc., Shiga, Japan). PCR was performed for this region using the following thermal cycling conditions: initial denaturation at 98 °C for 1 min followed by 30 cycles of 98 °C for 10 s, 55 °C for 30 s, and 72 °C for 5 min, and a final extension at 72 °C for 5 min. To select the appropriate restriction enzyme with distinguishable restriction sites in the sequence, inserted gene cassettes of the class 1 integron of the representative isolate were selected for sequencing. PCR products of the remaining isolates were digested using the restriction enzyme, and the RFLP pattern was confirmed using electrophoresis.

### 4.5. MLST Analysis

MLST analysis using seven housekeeping genes (*aroC*, *dnaN*, *hemD*, *hisD*, *purE*, *sucA*, and *thrA*) [25] was performed for the representative isolates. Each gene was amplified with PCR using Go Taq Green Master Mix (Promega K. K.), according to the methods described in a previous study [41]. The primer sequences for PCR and/or DNA sequencing and the expected PCR product sizes are listed in Appendix A. PCR products were purified and sequenced as described above. The sequences were submitted to the Salmonella Genome Databases [23], and the existing allele types were assigned to each of the seven loci. STs were assigned based on a combination of allele types. For comparison with STs for isolates in this study, the STs and the geographical distribution of *S.* Schwarzengrund strains were obtained from the Salmonella Genome Databases [23] and EnteroBase [24]. To reveal the relationship between STs, improved minimal spanning tree based on the isolates that were typed using MLST and registered in Enterobase were generated using GrapeTree, a tree-visualization program [42], within EnteroBase.

### 4.6. WGS

WGS was performed to detect antimicrobial resistance genes and STs. Representative isolates were selected for the WGS analysis. Total DNA was extracted and purified according to the methods described in a previous report [43]. Purified DNA was sequenced on an iSeq System (Illumina K. K., Tokyo, Japan) according to the manufacturer’s instructions using the Nextera XT Library Prep Kit (Illumina K. K.). The pre-assembled sequence reads from each isolate were submitted to ResFinder [44] for identification of acquired antimicrobial resistance genes and MLST for the determination of STs. To detect acquired antimicrobial resistance genes, the threshold for %ID and minimum length was set at 90% and 60%, respectively. The antimicrobial resistance gene detected by ResFinder was also tested for other isolates with PCR using Go Taq Green Master Mix (Promega K. K.).

### 4.7. Statistical Analysis

The proportion of resistant isolates was compared using the chi-squared test or Fisher’s exact test. *p*-values < 0.05 are considered significant. Statistical analyses were performed using R version 3.4.2.

## 5. Conclusions

*Salmonella enterica* serovar Schwarzengrund that spread in Japan showed different antimicrobial resistance patterns from those that have spread internationally. Seventy-eight isolates harbored a 1.0-kb class 1 integron containing *aadA1*, which is also different from that reported in globally distributed strains (1.9 kb, containing the *dfrA12-aadA2* gene cassette). *S*. Schwarzengrund ST241 may have contaminated broiler chickens in West Japan in 2008 and spread to East Japan. Hygiene measures of broiler farms need to be strengthened. The increase and decrease in antimicrobial use in broiler chickens were thought to contribute to a significant increase and decrease in antimicrobial resistance among *S*. Schwarzengrund. Prudent use of antimicrobial agents in veterinary medicine can reduce the number of antimicrobial-resistant bacteria.

## Figures and Tables

**Figure 1 antibiotics-10-01336-f001:**
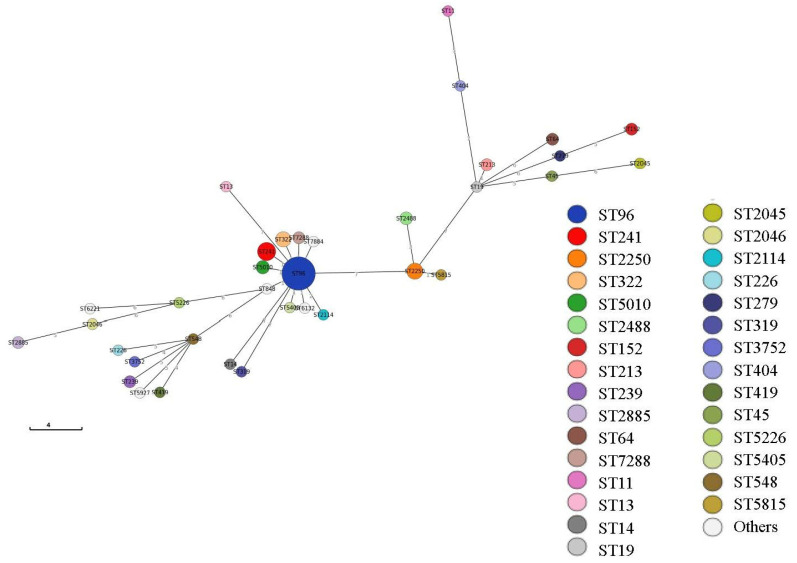
Improved minimal spanning tree called MSTree V2 based on 1833 *Salmonella enterica* serovar Schwarzengrund isolates that were typed via Achtman 7 gene multilocus sequence type and registered in EnteroBase [24] on 5 March 2021.

**Table 1 antibiotics-10-01336-t001:** Antimicrobial resistance of *Salmonella enterica* serovar Schwarzengrund isolates from chicken meat in Japan.

Antimicrobial Agents	Breakpoint	No. of Resistant Isolates (%)	
	(μg/mL)	2008 (n = 37)	2015–2019 (n = 87)
Ampicillin	32	0	0
Cefazolin	8	0	0
Streptomycin or Dihydrostreptomycin ^(a)^	16	37 (100%) ^(b)^	41 (47.1%) ^(b)^
Gentamicin	16	0	0
Kanamycin	64	19 (51.4%) ^(c)^	78 (89.7%) ^(c)^
Tetracycline	16	37 (100%) ^(d)^	41 (47.1%) ^(d)^
Chloramphenicol	32	0	0
Nalidixic acid	32	0^e)^	8 (9.2%) ^(e)^
Ciprofloxacin	1	0	0
Trimethoprim	16	21 (56.8%) ^(f)^	14 (16.1%) ^(f)^
Trimethoprim-sulfamethoxazole	4/76	26 (70.3%) ^(g)^	39 (44.8%) ^(g)^

^(a)^ Resistance to dihydrostreptomycin was tested for 29 isolates obtained from samples produced in 2008 [22]. Resistance to streptomycin was tested for the remaining isolates. ^(b, c, d, f)^ *p* <0.001; ^(e)^ *p* = 0.1038; ^(g)^ *p* = 0.009.

**Table 2 antibiotics-10-01336-t002:** Antimicrobial resistance genes present in *Salmonella enterica* serovar Schwarzengrund isolated from retail chicken meat in Japan.

Antimicrobial Resistance Gene	Class 1 Integron (Gene Casettes)	Antimicrobial Resistance	2008 ^(a)^	2015–2019			
			West Japan ^(b)^	West Japan	East Japan	Domestic ^(c)^	Sub-Total
*aac(6’)-Iaa-aadA1-tetA-sul1-dfrA14-aphA1*	1.0 kb (*aadA1*)	STR-TET-KAN	1				1
		STR-TET-KAN-NAL-SXT		2		3	5
		STR-TET-KAN-NAL-TMP-SXT		1		1	2
		STR-TET-KAN-SXT	1	4	9	6	20
		STR-TET-KAN-TMP-SXT	7		5	1	13
*aac(6’)-Iaa-aadA1-tetA-sul1-dfrA14*	1.0 kb (*aadA1*)	STR-TET-SXT	5		1	1	7
		STR-TET-TMP-SXT	13				13
*aac(6’)-Iaa-aadA1-tetA-sul1-aphA1*	1.0 kb (*aadA1*)	STR-TET-KAN	9		1	3	13
		STR-TET-KAN-TMP	1				1
*aac(6’)-Iaa-aadA1-tetA-sul1*	1.0 kb (*aadA1*)	STR-TET			2	1	3
*aac(6’)-Iaa-strA*/*strB-sul2-dfrA14-aphA1*	Not determined	KAN-TMP-SXT			3	2	5
*aac(6’)-Iaa-aphA1*	Not determined	KAN		1	24	8	33
		KAN-NAL			1		1
*aac(6’)-Iaa-dfrA14-aphA1*	Not determined	KAN-TMP			1		1
		KAN		1		1	2
*aac(6’)-Iaa*	Not determined	susceptible		1	2	1	4
Total			37	10	49	28	124

STR, streptomycin; TET, tetracycline; KAN, kanamycin; NAL, nalidixic acid; TMP, trimethoprim; SXT, trimethoprim-sulfamethoxazole. Minimum inhibitory concentrations for dihydrostreptomycin were determined for isolates obtained in 2008 in a previous study [22], but not for STR. Therefore, dihydrostreptomycin was transcribed as STR, for ease of comparison. ^(a)^ Years of isolation, ^(b)^ the product region for samples, ^(c)^ the product district could not be identified.

## Data Availability

The data presented in this study are available on request from the corresponding author.

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
