# Peer review of "Molecular Characteristics and Antimicrobial Resistance of Salmonella enterica Serovar Schwarzengrund from Chicken Meat in Japan"

_antibiotics, 2021, doi:10.3390/antibiotics10111336_

Round 1

Reviewer 1 Report

Antibiotic resistant Salmonella are major foodborne health concern. The manuscript entitled” Molecular characteristics and antimicrobial resistance of Salmonella enterica serovar Schwarzengrund from chicken meat in 3 Japan” is very well written. Here the authors have successfully characterize S. enterica serovar Schwarzengrund from chicken meat in Japan along with determination of their resistance pattern. Advance molecular based approaches were used in this study including  whole genome sequencing to conform the characterization and support the claimed findings. However, I have few comments that need to be address properly:

Line 107, please submit those WGS in GenBank and add those accession. Number here.

Line 238: Any explanation/speculation why none of the isolates were resistant to AMP, GEN, CHL, or CIP.??

Line 278: Please mention what was the basis of selected specifically those 11 antimicrobial agents for MIC test.

Lone 305: What was the minor modifications in the multiplex PCR used to detect the resistance genes, also mentioned what we the PCR positive and negative control??

Line 310: what type of BLAST was that, n,x,p please clarify??

Line 341. under 4.5. WGS , please write how the sequence quality was checked?? 

Line 347: what was setup in ResFinder, default?? make it clear...

Figure S1, show the negative control in the gel??

General comment: Name the location of the company where applicable for example..

Illumina, ??? CA, USA??  [as done in line 345]

(Promega Corporation, location/country???)

Author Response

Dear Reviewer 1

Thank you very much for your helpful comments on our submission.

We have revised the manuscript according to your suggestions.

Although we confirmed one comment to register the sequences identified by WGS in the databank, it will take a long time. Instead, we have created a new Table S1 with the data on the sequences of antimicrobial resistance genes identified by ResFinder.

In order to create this table, we searched antimicrobial resistance genes through ResFinder again. As a result, one antimicrobial resistance gene that was not detected before was found in this analysis. This gene had been detected by PCR before, so the difference between the two analyses (PCR and ResFinder) has been resolved.

In addition, when rechecking our manuscript, we found several errors in the isolate code and corrected them.

Therefore, we hope that you understand that some of the results have been changed from the previous manuscript.

The responses to individual comments and the revisions are as follows.

Comments from Reviewer 1

Comment 1, Line 107, please submit those WGS in GenBank and add those accession. Number here.

Response 1, From our experiences, it takes three or four weeks to register the determined sequences in the data bank. For antimicrobial resistance genes detected by ResFinder, the Accession No. of the reference sequence and the number of matched sequences are shown in Table S1 of revised manuscript.

Comment 2, Line 238: Any explanation/speculation why none of the isolates were resistant to AMP, GEN, CHL, or CIP.??

Response 2, We think that our isolates in Japan were susceptible to these antimicrobials, because S. Schwarzengrund in Japan was different from the globally distributed S. Schwarzengrund. Then, we added “antimicrobial susceptibilities” in to the last sentence in this paragraph (Line 261).

Comment 3, Line 278: Please mention what was the basis of selected specifically those 11 antimicrobial agents for MIC test.

Response 3, We chosen these 11 antimicrobials according to Japanese Veterinary Antimicrobial Resistance Monitoring System. Therefore, we added this reason in line 298-299.

Comment 4, Line 305: What was the minor modifications in the multiplex PCR used to detect the resistance genes, also mentioned what we the PCR positive and negative control??

Response 4, We used different DNA polymerase with previous report. DNA polymerase used in our PCR was described (Line 327). In the previous report, one multiplex PCR reaction detected three kinds of resistance genes (tetA, tetB and tetC or sul1, sul2, and sul3). However, our multiplex PCR detected only two genes without tetC and sul3.

We added positive control isolate for each gene in Table S6. Although we used sterilized distilled water as a negative control. In case of multiplex PCR, the positive control of one gene (ex. tetA) also served as the negative control of the other gene (ex. tetB).

Comment 5, Line 310: what type of BLAST was that, n,x,p please clarify??

Response 5, we used Nucleotide BLAST (BLASTn), so we added it in a revised manuscript (Line 98, 334).

Comment 6, Line 341. under 4.5. WGS, please write how the sequence quality was checked??

Response 6, ResFinder can analyze not only assembled genome/contigs but also single or paired end reads to detect antimicrobial resistance genes. Therefore, we analyzed pre-assembled reads and identify antimicrobial resistance genes by ResFinder without quality check.

The coverage of each gene is displayed when antimicrobial resistance gene is detected by ResFinder. The lowest value was 97.25 for tetA in Sal_278, and most of the genes had the highest value of 100. The range of coverage for all detected genes is shown in Table S1, which was newly created in the revised manuscript.

Comment 7, Line 347: what was setup in ResFinder, default?? make it clear...

Response 7, we detect acquired antimicrobial resistance genes as default set up. We added “To detect acquired antimicrobial resistance genes, threshold for %ID and minimum length was set at 90% and 60%, respectively.” (Line 383-384)

Comment 8, Figure S1, show the negative control in the gel??

Response 8, I may not fully understand the meaning of the negative control you pointed out, Figure S1 does not show integron including other antimicrobial gene(s) than aadA1. However, we could confirm that the RFLP pattern was different among integrons including aadA1 or aadA2. The isolates harbored integron with aadA2 were Salmonella Infantis but not Salmonella Schwarzengrund. Therefore, we did not add this isolate with aadA2 in Figure S1.

Comment 9, General comment: Name the location of the company where applicable for example..

Illumina, ??? CA, USA??  [as done in line 345]

(Promega Corporation, location/country???)

Response 9, We have confirmed that the location of each reagent manufacturer is stated in the text. If the company name is mentioned for the second time in the manuscript, the location is not stated. In the previous manuscript, the company name and location of the head office were described, but the company for which a corporation has been established in Japan has been changed to a Japanese corporation.

Reviewer 2 Report

Comments:

This is an interesting manuscript studying the origins, antimicrobial resistance and genotypes of S. Schwarzengrund in Japan. There are, however, few issues to address, detailed below:

  1. It would be good if the authors mentioned why they did not use dihydrostreptomycin in this study?
  2. Why did the authors performed WGS for only 5 isolates?

Minor comments:

  1. Line 65 - Replace "was" with "were".
  2. Line 132 - Replace "Therfore" with "Therefore"
  3. Line 205 - Authors should clearly indicate if "Isolates from OR in the UK"! Correct sentence.
  4. Line 221 - Correct sentence: ...also matched "with ST241". Replace "was" with "were"
  5. Line 341 and 351 - Label the sections correctly with "4.6" and "4.7" respectively.
  6. Figure S1 - Could the authors label the lanes and have they included necessary controls?

Author Response

Dear Reviewer 2

Thank you very much for your helpful comments on our submission.

We have revised the manuscript according to your suggestions.

Although we confirmed one comment from Reviewer 1 to register the sequences identified by WGS in the databank, it will take a long time. Instead, we have created a new Table S1 with the data on the sequences of antimicrobial resistance genes identified by ResFinder.

In order to create this table, we searched antimicrobial resistance genes through ResFinder again. As a result, one antimicrobial resistance gene that was not detected before was found in this analysis. This gene had been detected by PCR before, so the difference between the two analyses (PCR and ResFinder) has been resolved.

In addition, when rechecking our manuscript, we found several errors in the isolate code and corrected them.

Therefore, we hope that you understand that some of the results have been changed from the previous manuscript.

The responses to individual comments and the revisions are as follows.

Comments from Reviewer 2

Comment 1.        It would be good if the authors mentioned why they did not use dihydrostreptomycin in this study?

Response 1, We chosen these 11 antimicrobials according to Japanese Veterinary Antimicrobial Resistance Monitoring System (JVARM). We added this reason in line 298. Dihydrostreptomycin is an antibacterial agent often used in Japan for animals. However, streptomycin is used in human medicine in Japan. We believe that streptomycin is better when compared to antimicrobial resistance in Salmonella with animals or foods and human patients.

Comment 2,        Why did the authors perform WGS for only 5 isolates?

Response 2, After we performed WGS for 5 isolates, the antimicrobial resistance gene detected by ResFinder was also tested for other isolates with PCR.

As results with PCR, we thought that the antimicrobial resistance gene that causes antimicrobial resistance could be determined, and WGS was not performed on more isolates.

Minor comments:

  1. Line 65 - Replace "was" with "were".

Response M1, we modified it (Line 65).

  1. Line 132 - Replace "Therfore" with "Therefore"

Response M2, we modified it (Line 141).

  1. Line 205 - Authors should clearly indicate if "Isolates from OR in the UK"! Correct sentence.

Response M3, the original samples (human patient, chicken meat or another) for the isolate was not described in the previous report. Therefore, I revised this sentence to “Isolates obtained in the UK in 1998, Taiwan and Denmark.” (Line 218)

  1. Line 221 - Correct sentence: ...also matched "with ST241". Replace "was" with "were"

Response M4, we modified them (Line 237).

  1. Line 341 and 351 - Label the sections correctly with "4.6" and "4.7" respectively.

Response M5, we corrected it (Line 364 and 387).

  1. Figure S1 - Could the authors label the lanes and have they included necessary controls?

Response M6, I am so sorry Figure S1 did not include the label for each lane. We added the label in revised Figure S1.

Figure S1 does not show integron including other antimicrobial gene(s) than aadA1. However, we could confirm that the RFLP pattern was different among integrons including aadA1 or aadA2. The isolates harbored integron with aadA2 were Salmonella Infantis but not Salmonella Schwarzengrund. Therefore, we did not add this isolate with aadA2 in Figure S1.